# Effectiveness of the Mindfulness-Based Social–Emotional Growth (MSEG) Program in Enhancing Mental Health of Elementary School Students in Korea

**DOI:** 10.3390/bs15030315

**Published:** 2025-03-05

**Authors:** Jongtae Kim, Seonyeop Kim, Misan W. D. Kim, Yong-Han Park, Kanguk Lee, Dong Sun Chung, Youn Hee Kim, Yong-Sil Kweon, Duk-Soo Moon, Hwa-Young Lee, June Sung Park, Yun Hyoung Kang, Seunghee Won, Young Sook Kwack

**Affiliations:** 1Department of Psychiatry, College of Medicine, The Catholic University of Korea, Seoul 06591, Republic of Korea; jongtaekim3@gmail.com (J.K.); yskwn@catholic.ac.kr (Y.-S.K.); 2Gimpo Shinpoong Elementary School, Gimpo 10113, Republic of Korea; bright110471@gmail.com; 3KAIST Center for Contemplative Science, Korea Advanced Institute of Science and Technology, Daejeon 34141, Republic of Korea; herenow365@gmail.com (M.W.D.K.); happysmile@kaist.ac.kr (Y.H.K.); 4Park’s Psychiatric Clinic, Jeju 63591, Republic of Korea; npdrs@hanmail.net; 5Department of Psychiatry, College of Medicine, Kangwon National University, Chuncheon 24341, Republic of Korea; kuleemd@kangwon.ac.kr; 6W Psychiatric Clinic, Seongnam 13524, Republic of Korea; dschungh@gmail.com; 7Department of Psychiatry, School of Medicine, Jeju National University, Jeju 63241, Republic of Korea; dr.moon@daum.net; 8Department of Psychiatry, Cheonan Hospital, College of Medicine, Soonchunhyang University, Cheonan 31151, Republic of Korea; leehway@gmail.com; 9Do Dream Clinic, Bucheon 14537, Republic of Korea; drjune6th@empas.com; 10Doctor’s Psychiatric Clinic, Seoul 06110, Republic of Korea; famolovely@gmail.com; 11Department of Psychiatry, School of Medicine, Kyungpook National University, Daegu 41944, Republic of Korea; 12WEE Center, Kyungpook National University Chilgok Hospital, Daegu 41404, Republic of Korea; 13National Center for Mental Health, Seoul 04933, Republic of Korea

**Keywords:** mental health promotion, social–emotional competency, mindfulness, compassion, elementary school, children

## Abstract

Since the COVID-19 pandemic, mental health challenges among children and adolescents in South Korea have intensified with rising rates of depression and suicide. Proactive interventions focusing on mental well-being are needed to address this critical issue. This study examines the effectiveness of the Mindfulness-based Social–Emotional Growth (MSEG) program for enhancing the mental health of elementary school students in Korea. This quasi-experimental study evaluated the impact of the MSEG program on students at an elementary school in Gyeonggi-do, South Korea. A total of 70 students (35 from lower grades [grades 1–3] and 35 from upper grades [grades 4–6]) participated in the 12-week intervention group, while 72 students were in the control group. Pre- and post-intervention assessments measured social–emotional, mindfulness, and clinical domains. The findings revealed significant improvements in emotional regulation across all grades. Lower-grade students showed substantially reduced anxiety and depression, while upper-grade students demonstrated enhanced resilience compared to the control group. These results indicate the potential of the MSEG program to promote early mental health when integrated into elementary school curricula. Further research is needed to adapt the program to diverse educational settings and optimize its effectiveness and impact.

## 1. Introduction

Child and adolescent mental health is a significant public health crisis in South Korea. Following the COVID-19 pandemic, mental health risks have substantially increased in this population ([29]; [45]). According to the National Health Insurance Service, 37,386 children and adolescents aged 6 to 17 received treatment for depression in 2022, which is approximately 1.6 times higher than the 23,347 reported in 2018, before the pandemic ([29]). Additionally, [45] ([45]) reported that the adolescent suicide rate was 7.1 per 100,000 individuals in 2021, representing a 10.1% increase from the previous year. Mental health issues in children and adolescents can significantly affect learning and interpersonal relationships and often lead to adult mental health problems if not treated early, underscoring the importance of early intervention ([18]; [22]; [27]).

Promoting child and adolescent mental health is essential for addressing psychological challenges early and minimizing their long-term effects ([42]). This proactive and demographic-centric approach focuses on enhancing psychological well-being, resilience, and coping mechanisms while fostering supportive environments ([42]). Due to their developmental phase, children and adolescents often experience volatile emotions that can escalate into disorders such as depression and anxiety ([14]). Early approaches that enhance social–emotional skills, such as emotional regulation, relationship management, and responsible behavior, can prevent the exacerbation of these issues ([50]). Promoting mental health in adolescence is widely regarded as a highly effective strategy to preempt both the immediate and long-term development of mental disorders ([42]).

Schools are particularly effective settings for mental health promotion among Korean children and adolescents, given the country’s high enrollment rates, which were 100% in elementary schools and 95.8% in middle schools as of 2024 ([46]). These figures highlight the potential for school-based interventions to reach most of this population. Universal implementation of such programs within schools can help reduce the stigma surrounding mental health issues for students, parents, and educators alike, creating a more supportive environment for seeking assistance ([16]). Furthermore, mental health promotion in schools is cost-effective, reinforcing the suitability of schools as key settings for these initiatives ([1]).

In a Western context, endeavors to improve students’ social–emotional competency via school-based mental health interventions have evolved ([9]). Early initiatives prioritized character-building education to promote moral development. Over time, the approach shifted toward Social Emotional Learning (SEL), which aims to cultivate five interrelated competencies that are crucial for thriving in social settings: self-awareness, self-management, social awareness, relationship skills, and responsible decision-making. However, character education faced practical hurdles stemming from its abstract, moral values-centric approach, while SEL lacked a comprehensive model for ideal human behavior, leading to their complementary convergence. Recently, focus has turned to mindfulness training ([10]; [11]). Whereas character education and SEL hinge on cognitive learning, mindfulness entails internalized embodiment training to foster self-regulation, flexibility, and openness ([11]). The prevailing global trend in school-based mental health promotion advocates integrating SEL with mindfulness training, as exemplified by initiatives such as Social, Emotional, and Ethical (SEE) learning originating in the United States ([3]).

In Korea, school-based mental health initiatives predominantly focus on identifying and addressing high-risk students rather than universally promoting mental well-being ([4]; [6]). A survey of key mental health professionals, including health and counseling educators in elementary, middle, and high schools, revealed that school-based mental health programs typically focus on identifying high-risk students and offering interventions such as counseling. Furthermore, other mental health initiatives are less active and underutilized ([4]). Character education aimed at enhancing social–emotional competencies among general students started in South Korea in the 1990s, which was later than in Western societies. Although character education and SEL have been the main approaches according to comprehensive plans, mindfulness/compassion-based training has recently begun to gain ground. Nevertheless, such programs often proceed independently or in smaller-scale combined formats ([53]). Most recently, a short-term pilot study has been reported on the SEE Learning program’s effectiveness ([35]).

Multiple barriers hinder the large-scale adoption of mindfulness-based or mindfulness-integrated SEL programs in South Korea, including a lack of systematic structure in leadership and implementation methods, limited number of qualified instructors, and tendency to use adult-oriented content that does not fully consider children’s cognitive development. Moreover, international programs are sometimes used without appropriate cultural adjustments, and many rely heavily on an instructor’s personal experience rather than systematic design. These factors have led to a lack of rigorous evaluations of program effectiveness ([53]). Given the worsening mental health status of children and adolescents, mental health promotion initiatives adapted to the Korean context must be systematically created and assessed.

In response to global trends and the deteriorating mental health of Korean children and adolescents, our research team developed the Mindfulness-based Social–Emotional Growth (MSEG) program in 2023, specifically designed for elementary school students. Based on an understanding of children’s developmental traits and cognitive abilities, the MSEG program aims to enhance social–emotional capabilities through mindfulness and compassion training while teaching skills to identify and commit to universal values. This study aimed to assess the preliminary effectiveness of the MSEG program among Korean elementary school students.

## 2. Theoretical Background

The MSEG program aims to promote mental health and resilience in children and adolescents by fostering kindness toward oneself and others, self-regulation, self-awareness, emotional stability, gratitude, and an accepting attitude. Additionally, it seeks to cultivate an understanding of others’ perspectives and a sense of connection with others, nature, and the world. This program for elementary students was developed based on the core mindfulness principles of presence–moment awareness, non-judgment, acceptance, non-striving, patience, beginner’s mind, trust, letting go, gratitude, and universality ([21]; [41]) and incorporated compassion, which is defined by elements such as recognizing suffering, understanding the universality of suffering, emotional resonance, tolerating uncomfortable feelings, and the motivation to act to alleviate suffering ([47]). The core objectives of the MSEG program are as follows:Mindfulness: Awareness of the present moment as it is, with intentional and non-judgmental attention to the object of focus.Acceptance and Presence: Staying in the present moment and accepting the experience of the current moment.Managing the Self: Self-compassion, self-regulation, a sense of oneself as a whole, and self-esteem.Relationship with Others: Compassion for others and effective communication.Affirmative Attitude: Kindness and gratitude.Perspective Taking: Understanding others’ perspectives and adopting an observer’s perspective on oneself.Vital and Meaningful Life: Awareness of personal values and committed action.Experience Common Humanity and Connection with the World: Realizing the oneness of the world (non-dualism) and fostering a nature-friendly attitude.

Mindfulness is one wing of a bird and the most essential practice in the MSEG program. Globally, numerous school-based mindfulness programs have been developed and implemented. The primary effects of mindfulness training for adolescents include increased sensory awareness, greater cognitive control, enhanced regulation of emotions, acceptance of transient thoughts and feelings, and the capacity to regulate attention ([44]). Recent meta-analyses on school-based mindfulness programs have indicated their effectiveness in enhancing prosocial behavior, resilience, executive function, and attention, as well as in reducing anxiety, attention problems/ADHD behaviors, and conduct issues ([39]).

Compassion is another wing of the bird, representing an essential attitude toward oneself and others. Mindfulness primarily focuses on an individual’s thoughts, feelings, and sensations, whereas self-compassion centers on personal suffering, making it a more effective meditation practice in negative situations. The positive effects of self-compassion practice include fostering a healthy attitude toward oneself during feelings of inadequacy or crisis, adopting a non-judgmental and open stance toward personal or others’ suffering with warmth, and reducing judgment, isolation, and emotional overwhelm while promoting a more positive perspective. Several programs designed to enhance compassion have been developed and implemented in various forms worldwide, with their effectiveness well-documented. A meta-analysis was performed on the effects of self-compassion and found that it reduces negative mental states such as anxiety, depression, stress, thought suppression, rumination, and shame while enhancing positive mental states, including life satisfaction, happiness, a sense of connection, self-confidence, self-assurance, and optimism ([54]).

The two wings of the MSEG program, mindfulness and compassion, are interconnected and aligned with the CASEL (Collaborative for Academic, Social, and Emotional Learning) framework, which defines five core SEL competencies ([32]). These competencies include recognizing one’s emotions and thoughts (self-awareness), regulating emotions and behaviors (self-management), understanding others’ perspectives (social awareness), establishing healthy relationships (relationship skills), and making responsible choices (responsible decision-making).

The MSEG program integrates mindfulness practices—including breathing exercises, body awareness, and compassion training—to support self-regulation and coping skills. Rather than functioning as a separate component, mindfulness enhances SEL instruction by providing practical strategies for applying SEL skills. For example, mindful breathing can aid in managing anxiety (self-management), while mindful listening fosters stronger interpersonal connections (relationship skills). By aligning its curriculum with these principles, the MSEG program ensures that students develop well-rounded skills essential for mental health and effective social interactions. This alignment means that MSEG sessions explicitly teach skills such as emotional recognition, impulse control, empathy, and collaborative problem-solving, reinforcing CASEL’s model and supporting the program’s holistic approach to social–emotional development. Recent evaluations of the effectiveness of the SEE Learning short-term program, a mindfulness- and compassion-based SEL program, revealed statistically significant improvements in resilience, self-efficacy, tolerance of negative affect, positive support relations, sense of control, and spontaneity. Additionally, significant improvements were observed in social and emotional competencies, including emotional regulation, social skills, empathy, and social tendencies ([35]).

This integration is intentional: while SEL curricula define the skills students need to develop, mindfulness offers a practical method for cultivating those skills through experiential learning and reflection. By grounding its approach in these established theories, MSEG is well-positioned to foster social–emotional growth and mental well-being in a developmentally appropriate and evidence-based manner.

## 3. Materials and Methods

The Korean Academy of Meditation in Medicine and the Korea Advanced Institute of Science & Technology (KAIST) Center for Contemplative Science, in collaboration, conducted the ‘School Mindfulness Science Program Project’ from July 2021 to June 2024, aimed at enhancing the socio-emotional competency of children and adolescents. The instructor training and effectiveness study were entrusted to the Wee Center at Chilgok Kyungpook National University Hospital and were conducted with the approval of the institution. This study was a retrospective analysis conducted on data published in 2024.

### 3.1. Study Design

The study employed a quasi-experimental design to compare outcomes between participants engaged in the 12-week MSEG program and those in the control group who maintained regular school routines without additional intervention. Outcome measures were assessed at two time points: baseline (Week 0) and program completion (Week 12).

### 3.2. Procedure

The research was conducted between September 2023 and December 2023 in two public elementary schools in Gyeonggi-do, Korea. Permission was sought from instructors, parents, and students, with instructors receiving orientation sessions and parents receiving invitation letters that contained detailed information. Students were also briefed on the research project. All study methods were approved by the School and District Education Office’s Research Committee, and the research process was conducted under the monitoring of the respective institution.

The instructional team comprised a practicing teacher with over a decade of teaching experience, a specialist in mindfulness, and another experienced teacher with expertise in mindfulness. All instructors completed the foundational and advanced modules of the MSEG curriculum before program implementation. These training modules followed a comprehensive process in which instructors, in collaboration with the MSEG development team, thoroughly reviewed the entire program from start to finish, participated in live demonstrations, and received detailed feedback. Furthermore, the instructors and MSEG developers held three peer supervision/meeting sessions before starting the program and another six during its execution. These sessions aimed to deepen the instructors’ comprehension of the program and ensure its quality.

### 3.3. Participants

The experimental group included 70 participants, comprising 35 from lower grades (grades 1–3) and 35 from upper grades (grades 4–6). Both the children and their parents provided signed informed consent. Eligibility criteria required participants to demonstrate a willingness to engage in the program, as well as sufficient cognitive and linguistic abilities to comprehend and interact with the content. Exclusion criteria included severe cognitive, developmental, or neurological conditions (e.g., significant intellectual disability, severe autism spectrum disorder) that could hinder participation, along with concurrent involvement in other psychological or mindfulness-based interventions. The participants were recruited separately for lower and upper grades because two versions of the MSEG program were designed with each tailored to a different age group. Most participants had no prior mindfulness training experience, and their parents sought mindfulness training to help their children manage emotional challenges independently. The control group included 72 participants (36 from lower grades and 36 from upper grades) recruited through school advertisements.

In the lower-grade cohort, the experimental group had 35 participants (45.7% male and 54.3% female), whereas the control group had 36 participants (50% male and 50% female). No significant gender differences were observed between groups (χ^2^, *p* = 0.72). In the upper-grade cohort, the experimental group included 35 participants (60% male and 40% female), and the control group had 36 participants (63.9% male and 36.1% female). These groups also showed no significant gender differences (χ^2^, *p* = 0.74).

### 3.4. Program

MSEG sessions were conducted for 40 min each week over 12 weeks. The program was designed for lower and upper elementary school cohorts to accommodate the developmental differences between younger and older students. The program’s development process and characteristics are outlined below.

#### 3.4.1. Development Process

A Task Force Team (TFT) was formed with seven individuals: the director and a researcher from the KAIST Center for Contemplative Science, and five psychiatrists specialized in children and adolescents. The TFT developed the MSEG program for elementary students over three years, from September 2020 to September 2023. The TFT conducted a literature review and engaged in various mindfulness practices to establish the program’s direction, objectives, and goals. The team’s refined direction for program development was as follows: First, the program is grounded in neuroscience and considers age, developmental stages, and cognitive levels, ensuring that explanations use simple, easily understandable language for children. To overcome linguistic limitations, symbols and metaphors are incorporated. Additionally, for upper elementary students, the ACT (acceptance and commitment therapy) Matrix is utilized to support value-setting and enhance psychological flexibility. Second, the program is designed to provide engaging and enjoyable experiences through intuitive activities that align with the developmental characteristics of children and adolescents. It encourages mindfulness in daily life by incorporating familiar and playful elements such as games and interactive exercises. Third, key meditation techniques are integrated with traditional Korean games, fostering a cultural connection and enhancing engagement. Fourth, all experiences are structured to progressively expand from the individual to others and ultimately to society (the system), reinforcing a sense of interconnectedness.

#### 3.4.2. Structure and Content

The MSEG program for elementary school students includes 12 sessions of 40 min each, aligning with the typical duration of elementary school classes in Korea. Each session is divided into three primary phases: introduction, main activity, and closing. The introduction phase provides a brief overview and initial mindfulness exercises, such as breathing or listening meditation. The main activity phase engages students in central activities to develop mindfulness, self-compassion, and social–emotional competencies. The closing phase involves reflective practices in which students articulate their experiences and plan how they will implement mindful practices in the upcoming week. Table 1 provides an overview of the learning topics, core competencies, and associated social–emotional skills covered in each session. For a detailed description of each session, including key activities and practical examples, refer to Appendix A.

To address students’ distinct developmental stages and cognitive capacities, the program is divided into two versions for lower and upper elementary grades, respectively. This differentiation is evident in the activities and content. For example, in the “Mindful Eating” session, lower elementary students engage in “Mindful Tangerine Eating,” focusing on the sensory experience. In contrast, upper elementary students participate in “Mindful Dried Persimmon Eating,” which includes sensory exploration and fosters gratitude for the creation process and effort involved. This adaptation ensures that the program aligns appropriately with the varying cognitive and emotional levels of students across different age groups.

### 3.5. Measures

The effectiveness of the MSEG program was evaluated across three key domains: social–emotional, mindfulness, and clinical. In the social–emotional domain, the assessment used the Social and Emotional Competencies Scale and Resilience Scale for Children (RSC). The latter focuses specifically on measuring resilience, a construct closely associated with social–emotional competence ([31]), but not directly addressed by the Social and Emotional Competencies Scale. The mindfulness domain was assessed using the Multifactor Mindfulness Scale for Adolescents (MMSA). For the clinical domain, the evaluation employed the Center for Epidemiological Studies Depression Scale for Children (CES-DC) and Revised Children’s Manifest Anxiety Scale (RCMAS).

The questionnaires were designed to be easily understood by elementary school students and were administered by program instructors. Each question was read aloud, and the students answered them on their own. Additional explanations were provided if students needed help understanding a question. Depending on their needs, students could take one or two 5-minute breaks. Participation was voluntary, and students could stop at any time. The survey took approximately 40–50 min. Upon completion of the questionnaires, the students’ well-being was subsequently evaluated.

#### 3.5.1. Social and Emotional Competencies Scale

We assessed students’ social and emotional competencies using the five subscales of the Social and Emotional Competences Scale integrated by [23] ([23]) and later refined by [5] ([5]). These subscales are as follows: emotional regulation (13 items, e.g., “I communicate my emotions clearly when I am anxious”), social skills (13 items, e.g., “I am the one who suggests what games to play when I hang out with my friends”), empathy (6 items, e.g., “I can understand a person’s emotions by observing their facial expressions”), social tendency (8 items, e.g., “I am accepted and welcomed by my friends without being ignored or rejected”), and interpersonal relations (7 items, e.g., “I happily greet my close friends when we meet”). Participants rated items on a five-point Likert scale, with higher scores denoting greater levels of emotional regulation, social skills, empathy, social tendency, and interpersonal relations. In our study, Cronbach’s α was 0.55 for emotional regulation, 0.64 for social skills, 0.72 for empathy, 0.45 for social tendency, and 0.60 for interpersonal relations.

#### 3.5.2. Resilience Scale for Children

Students’ resilience was assessed using the RSC developed by [20] ([20]) for 4–6 graders in South Korea. The RSC includes five subscales: self-efficacy (8 items, e.g., “I am capable of performing my tasks to the best of my ability”), tolerance of negative affect (8 items, e.g., “I am able to manage unpleasant emotions”), positive support relations (4 items, e.g., “I am able to find assistance when I need it”), power of control (6 items, e.g., “I can maintain focus on my interests, even when faced with interruptions”), and spontaneity (4 items, e.g., “I enjoy experimenting with new experiences”). The items were rated on a five-point Likert scale, with higher scores indicating greater resilience. In this study, Cronbach’s α was 0.87 for the total scale.

#### 3.5.3. Multifactor Mindfulness Scale for Adolescents 

Mindfulness was evaluated using the 19-item MMSA explicitly developed for Korean adolescents ([26]). This scale includes two primary dimensions: awareness and attitude. The awareness dimension comprises attention and awareness subscales, and the attitude dimension comprises subscales for openness, self-tolerance, and present orientation. The items were assessed on a five-point Likert scale, with higher scores indicating greater mindfulness. In this study, Cronbach’s α was 0.83 for the total scale.

#### 3.5.4. Center for Epidemiological Studies Depression Scale for Children

To assess depressive symptoms, we used the Korean translation of the CES-DC, originally developed by [52] ([52]). This self-administered tool comprises four subscales: depressive affect, positive affect, interpersonal problems, and somatic symptoms/behavioral decline. It includes 20 items, each rated on a four-point Likert scale, with total scores ranging from 0 to 60. Higher scores indicate more severe depression. The internal consistency of the Korean-translated version has been confirmed through research with South Korean elementary school students. Cronbach’s α was 0.90 in [25] ([25]) and 0.62 in this study.

#### 3.5.5. Revised Children’s Manifest Anxiety Scale

Anxiety symptoms were measured using the Korean translation of the RCMAS, initially developed by [2] ([2]) and revised by [40] ([40]). This self-report scale comprises 37 items, each requiring a “yes” or “no” response. Higher RCMAS scores reflect more significant anxiety symptoms. A previous study conducted with children and adolescents reported high internal consistency for the Korean-translated version, with a Cronbach’s α of 0.85 ([37]). In this study, Cronbach’s α was 0.80.

### 3.6. Statistical Analysis

Statistical analyses were performed using SPSS version 25. Independent sample *t*-tests were used to verify the initial homogeneity between groups. Repeated-measures analysis of variance (ANOVA) was used to evaluate program effectiveness. If significant pre-test differences were identified, analysis of covariance (ANCOVA) was employed to control for covariate effects. Separate analyses were conducted for lower- and upper-grade students, with a significance threshold of *p* < 0.05.

## 4. Results

### 4.1. Baseline Comparison

Table 2 and Table 3 present the outcomes of the independent sample *t*-tests conducted on all variables for the lower- and upper-grade cohorts before program implementation. Most scales and subscales did not show statistically significant differences between the experimental and control groups in either cohort. However, the lower-grade cohort exhibited a significant difference in the spontaneity subscale of the RSC (*p* = 0.01), indicating potential baseline heterogeneity in spontaneity. In contrast, the absence of significant differences across the measured variables in the upper-grade cohort suggests greater baseline homogeneity.

### 4.2. Post-Program Comparison

Table 4 and Table 5 show the repeated-measures ANOVA results for all variables in the lower- and upper-grade cohorts.

#### 4.2.1. Lower-Grade Students

Notably, in the lower grades, significant group-by-time interactions emerged in emotional regulation (F = 4.24, *p* < 0.05) in the social–emotional domain and anxiety (F = 4.41, *p* < 0.05) and depression (F = 4.50, *p* < 0.05) in the clinical domain, demonstrating the program’s effectiveness in the experimental group. However, aside from emotional regulation, other social–emotional and mindfulness domains showed no significant effects. The spontaneity subscale, which showed significant score differences in the lower grades before program implementation, was subjected to an ANCOVA. The analysis showed no statistically significant differences in program effects between groups (F = 2.46, *p* = 0.12), and the effect size (Cohen’s d = 0.29) indicated a small effect.

#### 4.2.2. Upper-Grade Students

In the upper-grade levels, significant group-by-time interactions were observed in the social–emotional domain in emotional regulation (F = 12.71, *p* < 0.01) and resilience (F = 4.24, *p* < 0.05), with significant effects found for the subscales of tolerance of negative affect (F = 4.01, *p* < 0.05) and positive support relations (F = 5.40, *p* < 0.05). These findings highlight the program’s effectiveness within the experimental group. In the other domains, the pre- and post-program changes revealed no significant differences between the experimental and control groups.

## 5. Discussion

This study investigated the effectiveness of the MSEG program, a 12-week curriculum designed to enhance the social–emotional competencies of elementary school students in Korea. The findings revealed significant gains across multiple domains. Specifically, both lower- and upper-grade students showed marked improvements in emotional regulation skills. Younger students demonstrated reduced anxiety and depression, while older students displayed heightened resilience. These initial results align with previous research supporting the beneficial effects of school-based SEL programs that incorporate mindfulness training ([7]; [35]; [43]).

The baseline analysis showed no significant differences between the experimental and control groups for most measures, supporting the reliability of the comparisons. A significant difference in the spontaneity subscale of the RSC was found in the lower-grade cohort, which was adjusted in a subsequent ANCOVA. The results were consistent before and after adjustment, with no significant differences in program effects between groups.

The post-program evaluation indicated improved emotional regulation and resilience and reduced anxiety and depressive symptoms. These changes were likely driven by the combined influence of mindfulness, compassion, and SEL elements in the MSEG program. Research supports the effectiveness of SEL interventions in enhancing students’ emotional skills, fostering prosocial behavior, and mitigating emotional distress ([8]). By targeting these areas, the program may have helped children develop the skills needed to manage their emotions and navigate social relationships more effectively, resulting in better mental health outcomes. Existing evidence aligns with these observations, indicating that SEL-based programs can bolster emotional regulation and resilience while mitigating anxiety and depression ([15]; [33]), which were the core domains showing measurable improvement in this study.

Mindfulness, a key component of the MSEG program, likely helped enhance positive mental health outcomes among the participants. [21] ([21]) defined mindfulness as a purposeful, non-judgmental awareness of present experiences, marked by intentional focus, continuous attentiveness to the present, and an attitude of openness and acceptance toward one’s thoughts and sensations. A key therapeutic mechanism of mindfulness is the exposure effect ([44]). Through focused meditation, individuals learn to relax while confronting anxiety-inducing situations, thereby helping them remain calm and gradually become desensitized to these triggers. Acceptance is another mechanism. Rather than attempting to suppress or avoid negative thoughts and feelings, which can often exacerbate anxiety, mindfulness encourages open and non-judgmental acceptance of these experiences, leading to reduced anxiety over time ([44]). Research supports the benefits of mindfulness, showing a positive link between increased mindfulness and resilience ([55]). Additionally, findings from a meta-analysis indicated the potential of mindfulness practices as effective interventions to reduce anxiety and depressive symptoms ([48]). Studies have further suggested that mindfulness meditation can change various brain regions associated with emotional regulation, including the prefrontal areas, limbic structures, and striatum ([49]). Together, these mechanisms likely reinforced the MSEG program’s effectiveness in fostering mental well-being.

Compassion or Self-compassion, another major component of the MSEG program, is the practice of treating oneself with kindness during times of perceived failure or difficulty. It involves recognizing that such experiences are part of the common human condition and approaching distressing thoughts and emotions with balanced awareness ([36]). Research indicates a negative correlation between self-compassion and anxiety and depression and highlights its positive association with resilience and emotional regulation ([34]; [38]; [51]). Neuroscientific studies also support these benefits. [12] ([12]) demonstrated that self-compassion exercises activate the parasympathetic nervous system, which aids relaxation. Moreover, [28] ([28]) found that through empathy and building resilience, compassion training activates brain areas associated with pain processing. These insights align with the improvements observed among the MSEG program participants.

Despite the program’s positive outcomes, further investigation is needed to explain the variations across the measured areas and age groups. For example, no significant effects were found in the mindfulness domain for either cohort, which suggests the limitations of the measurement tool. Currently, no well-established instruments are available for assessing mindfulness in Korean elementary students. The MMSA scale, validated for adolescents and used in this study, may not be developmentally appropriate for younger children, possibly leading to inaccurate responses owing to the abstract nature of the questions. This issue with scale suitability may have also contributed to the differences observed between groups for the resilience scale, as significant findings were noted in the upper but not the lower grades. As the RSC has been specifically validated for upper elementary school students in Korea, it may not effectively assess resilience levels among younger children.

Developmental differences in cognitive and emotional capacities may also explain the variations in program effects. Lower-grade students (ages 7–9) appeared to benefit more from the program’s experiential learning methods than their upper-grade peers. Instructor observations indicated that these students exhibited greater curiosity about mindfulness and higher psychological flexibility compared to older counterparts ([24]). This aligns with research suggesting that younger children, with their present-focused and experiential mindset, engage more effectively in mindfulness practices ([13]). The consistency between empirical observations and existing research suggests that younger children’s natural inclination for direct, hands-on engagement with mindfulness—fueled by their openness and curiosity—may be a key mechanism through which the MSEG program more effectively alleviates anxiety and depressive symptoms in comparison to older students. However, owing to the complex interaction between the various components of the MSEG program and participants’ developmental stages, combined with the limitations of the quasi-experimental research design, fully understanding these differential effects is challenging. Further research is needed to elucidate the program’s varying effects across age groups.

This study had several limitations. First, reliance on self-reported data may have introduced response bias. MSEG program participants were aware of their involvement, which could have influenced their answers. Future research should adopt a multifaceted assessment approach, incorporating observational or behavioral measures to address this issue. Second, selection bias may have influenced the experimental group, as the recruitment process likely attracted students and parents with a preexisting interest in mindfulness-based programs. This bias could predispose participants to respond more positively, potentially overstating the program’s benefits. Future studies should employ systematic sampling methods to ensure a more representative participant pool, thereby enhancing sample representativeness and improving generalizability. Third, the lack of long-term follow-up limits our understanding of the MSEG program’s sustained effects. Fourth, the absence of random assignment undermines the validity of our findings, leading to ambiguity in the interpretation of the results. Moreover, the quasi-experimental design inherently limits causal inference due to potential confounding by unmeasured variables. Therefore, future studies should consider using randomized controlled trials or advanced statistical methods (e.g., propensity score matching) to strengthen causal claims.

Despite these limitations, this study is significant because it demonstrates the preliminary effectiveness of a well-designed mindfulness-based SEL program in a Korean elementary school setting. A notable theoretical contribution of this research is its demonstration of the cultural adaptability of mindfulness-integrated SEL frameworks. The successful implementation of the MSEG program in a Korean school setting reinforces the feasibility of combining SEL and mindfulness beyond Western contexts, suggesting that such integrated approaches can produce beneficial outcomes across diverse cultural environments. This finding aligns with the broader global movement advocating for the integration of SEL and mindfulness in education and further expands empirical support for this approach within an East Asian cultural context.

Several research areas require further exploration to facilitate the widespread implementation of the MSEG program. First, investigating the impact of the MSEG program on the mental health outcomes of the teachers who administer it is essential. In Korea, teachers’ mental health has notably worsened, which has been attributed to increasing conflicts with parents and significantly reduced authority. Teachers’ well-being is crucial for effectively integrating SEL into schools ([19]); therefore, their mental health must be prioritized to implement the MSEG program successfully. [30] ([30]) found that teachers participating in school-based mindfulness training experienced reduced burnout. If the MSEG program benefits the mental health of participating teachers, this could be a compelling argument supporting the broader adoption of the program. Second, future research should explore the effects of varying dosages and intensities to align program delivery with specific objectives. Although repetition and practice are critical for inducing neural changes and fostering sustainable mental and physical health habits ([17]), resource constraints in real-world settings require a balanced approach. Examining how different frequencies and intensities of program delivery influence students’ mental health outcomes could yield critical insights into optimizing program implementation in resource-limited contexts. Third, further research is needed to develop and validate age-appropriate assessment tools for evaluating mindfulness and SEL outcomes. While the current study utilized the MMSA, a validated measure for adolescents, this instrument may not be ideally suited for younger children due to their distinct cognitive and emotional developmental stages. Developing tailored measurement tools for each age group will improve the accuracy of evaluations and ensure that interventions are appropriately targeted.

## 6. Conclusions

The MSEG program, which integrates mindfulness and compassion training with the cultivation of universally applicable values, demonstrates promising potential for improving mental health outcomes among elementary school students. This study’s results highlight the importance of embedding such programs into school curricula to foster mental well-being from an early age. Aligning with the global trend of incorporating mindfulness into SEL, this study expands the empirical literature on mindfulness-based practices in schools. As the first systematic evaluation of the MSEG program, this study lays the foundation for future rigorous randomized controlled trials to further examine the program’s effectiveness.

## Figures and Tables

**Table 1 behavsci-15-00315-t001:** Overview of learning topics, key and related social–emotional competencies per session in the MSEG program.

Session	Title	Key Competencies	Related Social–Emotional Competencies
1	Healthy Brain and Mindfulness	Introduction to mindfulness and understanding basic concepts Understanding neuroplasticity	Self-awareness
2	Matters Important to Me	Identifying and practicing what is important for my body and mind	Self-awareness
3	Mindfulness through Movements of Five Forest Animal Friends	Recognizing, awakening, and stabilizing sensations within the body Recognizing, praising, and learning from each other’s strengths	Self-awareness
4	Mindfulness in Caring for the Body	Accepting your body as it is Awakening the five senses Learning relaxation techniques through play	Self-awareness
5	Practicing Mindful Breathing	Learning breathing techniques Understanding the difference between long and short breaths	Self-awareness, self-management
6	Mindful Eating	Experiencing and recognizing eating through the five senses	Self-awareness, self-management
7	Mindfulness of Emotions	Reading emotions experienced in daily life Connecting situations, thoughts, and emotions when experiencing difficult feelings and managing them	Self-awareness, self-management
8	Mindfulness of Gratitude and Kindness	Learning to appreciate oneself and others and to treat oneself with care Developing a positive attitude of gratitude toward oneself and others and valuing self-care	Self-awareness, self-management, and social awareness
9	Mindfulness of Considering Others’ Perspectives	Experiencing perspective shifts Experiencing that each person may have different thoughts, feelings, and needs	Self-awareness, self-management, social awareness, and relationship skills
10	Mindfulness in Social Interaction	Experiencing interconnectedness through acts of kindness toward others Developing habits of kindness and consideration	Self-awareness, self-management, social awareness, and relationship skills
11	Mindfulness in Connecting with Nature	Recognizing “Screen Time” and “Green Time” Identifying natural objects that resemble oneself and feeling a sense of connection with nature Exploring and practicing ways to connect with nature	Self-awareness, self-management, social awareness, relationship skills, and responsible decision-making
12	Mindfulness for a Harmonious World	Understanding our interconnectedness Practicing actions to create a better world	Self-awareness, self-management, social awareness, relationship skills, and responsible decision-making

**Table 2 behavsci-15-00315-t002:** Independent sample *t*-test results for lower-grade students before program implementation.

Domain			Control	Experimental	
			Mean (SD)	Mean (SD)	*p*
Emotional Regulation			46.94 (6.27)	46.08 (5.50)	0.54
Social Skill			46.22 (6.37)	45.25 (5.17)	0.49
Empathy Competency			22.97 (3.50)	22.45 (3.96)	0.56
Social Disposition			30.77 (3.89)	29.42 (4.69)	0.19
Interpersonal Competency			29.19 (3.52)	27.57 (4.02)	0.08
Resilience					
	Total Score		115.77 (17.51)	108.20 (14.47)	0.051
	Self-efficacy		30.50 (4.78)	28.62 (4.70)	0.10
	Tolerance of Negative Affect		30.47 (5.01)	28.85 (3.93)	0.14
	Positive Support Relationship		14.57 (2.83)	14.57 (2.45)	0.11
	Power of Control		23.52 (4.19)	22.40 (3.65)	0.23
	Spontaneity		15.69 (2.64)	13.74 (3.20)	0.01 *
Mindfulness					
	Total Score		72.25 (11.92)	70.77 (10.79)	0.59
	Attention		13.86 (2.75)	14.40 (2.86)	0.42
	Awareness		11.83 (2.23)	11.20 (2.04)	0.22
	Openness		19.25 (3.39)	18.62 (3.42)	0.45
	Self-tolerance		15.63 (3.86)	15.11 (3.11)	0.53
	Present Orientation		11.66 (2.46)	11.42 (2.52)	0.69
Anxiety			15.97 (6.60)	16.08 (6.28)	0.94
Depression			21.08 (7.76)	20.60 (7.08)	0.79

* *p* < 0.05; SD = standard deviation.

**Table 3 behavsci-15-00315-t003:** Independent sample *t*-test results for upper-grade students before program implementation.

Domain			Control	Experimental	
			Mean (SD)	Mean (SD)	*p*
Emotional Regulation			47.88 (5.20)	47.08 (5.04)	0.51
Social Skill			46.94 (8.19)	46.22 (6.21)	0.68
Empathy Competency			22.13 (4.70)	22.25 (4.21)	0.91
Social Disposition			31.27 (4.70)	30.14 (2.93)	0.23
Interpersonal Competency			28.3 (4.25)	28.05 (2.90)	0.78
Resilience					
	Total Score		114.30 (20.61)	109.57 (12.67)	0.25
	Self-efficacy		30.44 (5.86)	29.57 (5.06)	0.51
	Tolerance of Negative Affect		30.13 (5.55)	28.28 (3.41)	0.10
	Positive Support Relationship		15.36 (3.27)	14.94 (2.78)	0.57
	Power of Control		23.61 (4.13)	23.45 (2.77)	0.85
	Spontaneity		14.75 (3.81)	13.31 (2.92)	0.08
Mindfulness					
	Total Score		71.16 (11.70)	71.62 (6.90)	0.84
	Attention		13.50 (2.85)	14.42 (1.66)	0.10
	Awareness		11.97 (2.45)	11.62 (1.53)	0.48
	Openness		18.83 (3.62)	19.91 (3.14)	0.19
	Self-tolerance		15.50 (3.75)	14.82 (2.84)	0.40
	Present Orientation		11.36 (2.69)	10.82 (2.78)	0.42
Anxiety			16.63 (5.65)	16.00 (5.83)	0.64
Depression			21.77 (7.37)	19.80 (6.96)	0.25

SD = standard deviation.

**Table 4 behavsci-15-00315-t004:** Repeated-measures ANOVA results for program outcomes in lower-grade students.

Domain				MS	F	*p*
Emotional Regulation			group	13.27	0.26	0.61
			Time	202.22	11.17	0.001
			time × group	76.72	4.24	0.04 *
Social Skill			group	3.16	0.05	0.83
			Time	3.94	0.36	0.55
			time × group	15.78	1.42	0.24
Empathy Competency			group	1.07	0.05	0.83
			Time	2.37	0.35	0.56
			time × group	4.14	0.61	0.44
Social Disposition			group	14.47	0.50	0.48
			Time	7.53	1.05	0.31
			time × group	17.93	2.50	0.12
Interpersonal Competency			group	36.46	1.67	0.20
			Time	2.71	0.54	0.47
			time × group	13.19	2.63	0.11
Resilience **						
	Total Score		group	791.47	1.81	0.18
			Time	267.33	2.51	0.12
			time × group	289.42	2.71	0.10
	Self-efficacy		group	38.37	0.96	0.33
			Time	0.10	0.01	0.92
			time × group	24.55	2.67	0.11
	Tolerance of Negative Affect		group	28.52	1.00	0.32
			Time	37.23	3.09	0.08
			time × group	18.33	1.52	0.22
	Positive Support Relationship		group	17.10	1.68	0.20
			Time	11.45	3.28	0.08
			time × group	3.59	1.03	0.32
	Power of Control		group	15.79	0.61	0.44
			Time	17.93	3.92	0.05
			time × group	7.53	1.65	0.20
Mindfulness						
	Total Score		group	21.52	0.08	0.78
			Time	102.58	3.43	0.07
			time × group	17.39	0.58	0.45
	Attention		group	8.59	0.61	0.44
			Time	16.19	6.25	0.02
			time × group	0.08	0.03	0.86
	Awareness		group	7.30	0.87	0.35
			Time	1.99	0.98	0.33
			time × group	1.15	0.56	0.46
	Openness		group	9.02	0.42	0.52
			Time	4.10	0.93	0.34
			time × group	0.49	0.11	0.74
	Self-tolerance		group	3.04	0.15	0.71
			Time	7.32	3.66	0.06
			time × group	1.91	0.95	0.33
	Present Orientation		group	0.02	0.001	0.97
			Time	7.77	2.65	0.11
			time × group	1.69	0.58	0.45
Anxiety			group	41.29	0.57	0.45
			Time	95.06	8.30	0.01
			time × group	50.44	4.41	0.04 *
Depression			group	129.15	1.64	0.21
			Time	159.32	9.96	0.002
			time × group	71.99	4.50	0.04 *

* *p* < 0.05 in the time × group interaction; ** An ANCOVA was performed on the spontaneity subscale of resilience, which initially demonstrated significant differences among lower-grade participants before program implementation. However, the analysis revealed no statistically significant differences in the program’s effects between groups (F = 2.46, *p* = 0.12); MS = mean square.

**Table 5 behavsci-15-00315-t005:** Repeated-measures ANOVA results for program outcomes in upper-grade students.

Domain				MS	F	*p*
Emotional Regulation			group	60.53	1.23	0.27
			Time	108.45	8.73	0.004
			time × group	157.89	12.71	0.001 *
Social Skill			group	4.03	0.04	0.84
			Time	108.35	7.77	0.01
			time × group	39.34	2.82	0.10
Empathy Competency			group	7.30	0.21	0.65
			Time	5.69	1.13	0.29
			time × group	11.61	2.31	0.13
Social Disposition			group	30.24	1.06	0.31
			Time	38.78	7.31	0.01
			time × group	1.59	0.30	0.59
Interpersonal Competency			group	5.17	0.21	0.65
			Time	10.10	2.48	0.12
			time × group	0.63	0.16	0.70
Resilience						
	Total Score		group	112.53	0.22	0.64
			Time	1154.61	15.82	0.000
			time × group	309.63	4.24	0.04 *
	Self-efficacy		group	0.56	0.01	0.92
			Time	41.47	7.09	0.01
			time × group	19.84	3.39	0.07
	Tolerance of Negative Affect		group	23.72	0.63	0.43
			Time	83.71	8.81	0.004
			time × group	38.07	4.01	0.049 *
	Positive Support Relationship		group	0.75	0.05	0.82
			Time	34.07	16.36	0.000
			time × group	11.25	5.40	0.02 *
	Power of Control		group	0.83	0.04	0.85
			Time	23.12	6.53	0.01
			time × group	3.35	0.95	0.33
	Spontaneity		group	45.78	2.74	0.10
			Time	59.98	10.58	0.002
			time × group	3.19	0.56	0.46
Mindfulness						
	Total Score		group	45.78	0.24	0.63
			Time	286.37	8.10	0.01
			time × group	16.11	0.46	0.50
	Attention		group	30.92	2.55	0.12
			Time	4.06	1.18	0.28
			time × group	0.001	0.000	0.99
	Awareness		group	0.07	0.01	0.92
			Time	3.19	1.90	0.17
			time × group	3.19	1.90	0.17
	Openness		group	30.29	1.46	0.23
			Time	25.21	5.14	0.03
			time × group	0.88	0.18	0.67
	Self-tolerance		group	12.54	0.72	0.40
			Time	7.70	1.72	0.19
			time × group	0.21	0.05	0.83
	Present Orientation		group	0.25	0.02	0.88
			Time	28.35	7.37	0.01
			time × group	7.16	1.86	0.18
Anxiety			group	1.52	0.03	0.87
			Time	0.17	0.02	0.89
			time × group	25.41	3.05	0.09
Depression			group	18.55	1.01	0.32
			Time	3.96	0.22	0.64
			time × group	18.55	1.01	0.32

* *p* < 0.05 in the time × group interaction; MS = mean square.

## Data Availability

Subsets of data are available from the corresponding authors upon reasonable request.

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
