# Peer review of "Effectiveness of the Mindfulness-Based Social–Emotional Growth (MSEG) Program in Enhancing Mental Health of Elementary School Students in Korea"

_behavsci, 2025, doi:10.3390/bs15030315_

Round 1

Reviewer 1 Report

Comments and Suggestions for Authors

This manuscript presents an interesting and timely topic on the effectiveness of the mindfulness-based social-emotional growth program in enhancing mental health of elementary school students in Korea. However, it requires some revisions to improve clarity and flow.

Literature Review

  • While authors discussed approaches like SEL and mindfulness, the theoretical foundations for these approaches and how they specifically informed the development of the MSEG program could be made more explicit. Please include the key theories underpinning SEL and mindfulness to strengthen your theoretical rationale.

Method

  • Please provide a detailed description of the MSEG program. E.g., specific examples of key activities for each session or at least for each of the main components
  • Please provide inclusion and exclusion criteria used for participant selection.
  • Provide a brief description of MSEG training.

Discussions:

  • Does this study contribute to any specific theories related to child mental health, SEL, mindfulness, or cultural adaptation of interventions? If so, please state these theoretical contributions. Even if the theoretical contribution is incremental (e.g., further supporting the role of mindfulness in SEL in a non-Western context), making it explicit enhances the perceived significance.
  • Some interpretation can be directly linked back to your literature review. E.g., when discussing age differences, you may connect this to the instructor observations and the Goodman & Greenland (2009) study to build a stronger interpretive argument.

Author Response

We would like to thank the editor and reviewers for providing valuable comments to enhance the quality of the manuscript. We have revised the manuscript as follows

1. Literature Review

While authors discussed approaches like SEL and mindfulness, the theoretical foundations for these approaches and how they specifically informed the development of the MSEG program could be made more explicit. Please include the key theories underpinning SEL and mindfulness to strengthen your theoretical rationale.

-> We thank the reviewer for the valuable comment. In response, we have revised the manuscript by adding a new section, 2. Theoretical Background, which explicitly outlines the key theories underpinning SEL and mindfulness (Lines 130-206).

2. Method

Please provide a detailed description of the MSEG program. E.g., specific examples of key activities for each session or at least for each of the main components.

-> In response,

1) We have additionally refined the development direction (Lines 269-283)..

2) A detailed description of the MSEG program has been provided as Supplementary Material S1 (Lines 294-295).

3. Method

Please provide inclusion and exclusion criteria used for participant selection.

-> In response to the comment, we have added inclusion and exclusion criteria in the first paragraph of 3.3. Participants section (Lines 242-247).

4. Method

Provide a brief description of MSEG training

-> In response to the comment, we have added a brief description of MSEG training in the second paragraph of 3.2. Procedure section (Lines 232-235).

5. Discussions

Does this study contribute to any specific theories related to child mental health, SEL, mindfulness, or cultural adaptation of interventions? If so, please state these theoretical contributions. Even if the theoretical contribution is incremental (e.g., further supporting the role of mindfulness in SEL in a non-Western context), making it explicit enhances the perceived significance.

-> Thank you for your valuable comment. In our revised Discussion section, we have explicitly addressed our study’s theoretical contribution. Specifically, we have emphasized that the successful implementation of the MSEG program in a Korean school context supports the cultural adaptability of mindfulness-integrated SEL frameworks (Lines 529-536).

6. Discussions

Some interpretation can be directly linked back to your literature review. E.g., when discussing age differences, you may connect this to the instructor observations and the Goodman & Greenland (2009) study to build a stronger interpretive argument.

-> In response, we have revised the seventh paragraph of the Discussion section to explicitly integrate our empirical observations with the existing literature to reinforce the interpretive argument (Lines 502-506).

Reviewer 2 Report

Comments and Suggestions for Authors

Dear Authors,

I would like to provide a review and suggestions for improving this article. My focus is on identifying inconsistencies and inaccuracies, while also justifying the suggested points for improvement. Additionally, I will address the limitations of the study beyond those already mentioned in the article.

Overall, the article is well-structured and presents a research problem that is relevant not only in Korea but globally. The methodology is clearly described throughout the article; however, my comments will focus on specific methodological choices and the data analysis. Below are my recommendations:

Limitations:

  1. The quasi-experimental design has inherent limitations. I suggest providing a more detailed discussion of potential methodological biases and ways to address them in future research.
  2. While the article mentions that the MMSA is not suitable for younger children, this point should be further reinforced in the Discussion section. The same applies to:
  3. sample selection bias – The study sample is likely affected by recruitment bias, as it consists of students/parents who are already interested in mindfulness practices. This limitation should be explicitly acknowledged and discussed.

Data Analysis:

  1. In the ANCOVA analysis, I recommend including an effect size analysis (Cohen’s d) to provide a better understanding of the practical significance of the results.

Other Points:

  • Bibliographic References: The citations need to be revised to ensure consistency and adherence to a standardized format. Specifically, the use of DOIs and URLs should be uniform across all references.

Author Response

Title: Effectiveness of the Mindfulness-based Social-Emotional Growth (MSEG) Program in Enhancing Mental Health of Elementary School Students in Korea

We would like to thank the editor and reviewers for providing valuable comments to enhance the quality of the manuscript. We have revised the manuscript as follows.

1. Limitations

The quasi-experimental design has inherent limitations. I suggest providing a more detailed discussion of potential methodological biases and ways to address them in future research.

-> Thank you for your valuable feedback. We have revised the eighth paragraph of the Discussion section to incorporate your suggestion (Lines 523-526).

2. Limitations

While the article mentions that the MMSA is not suitable for younger children, this point should be further reinforced in the Discussion section. The same applies to:

-> We appreciate the comment. In response, we have revised the tenth paragraph of the Discussion section to more explicitly emphasize the limitations of using the MMSA for younger children. This addition reinforces the point that while the MMSA was used in our study, its applicability to younger children is limited, and future research should address this gap (Lines 553-559).

3. Limitations

sample selection bias – The study sample is likely affected by recruitment bias, as it consists of students/parents who are already interested in mindfulness practices. This limitation should be explicitly acknowledged and discussed.

-> In response, we have revised the eighth paragraph of the Discussion section to explicitly acknowledge and discuss the potential recruitment bias (Lines 515-520).

4. Data Analysis

In the ANCOVA analysis, I recommend including an effect size analysis (Cohen’s d) to provide a better understanding of the practical significance of the results.

-> We calculated the effect size (Cohen’s d = 0.29) to assess the practical significance of the findings and included this information into the results section (Line 410).

5. Other Points

Bibliographic References: The citations need to be revised to ensure consistency and adherence to a standardized format. Specifically, the use of DOIs and URLs should be uniform across all references.

-> We have revised the citations to ensure consistency and adherence to a standardized format, with a particular focus on the uniform use of DOIs and URLs across all references.
